# Characterization of Different Bentonites and Their Properties as a Protein-Fining Agent in Wine

Stephan Sommer *, Stella J. Sommer and Monica Gutierrez

Viticulture and Enology Research Center, California State University, 2360 E. Barstow Ave, Fresno, CA 93740, USA; stellasommer@csufresno.edu (S.J.S.); monicagutierrez1995@mail.fresnostate.edu (M.G.)
* Correspondence: ssommer@csufresno.edu

**Abstract:** Bentonite is a natural clay that is used in different industries as a desiccant, ion-exchange material, or additive to remove impurities. For example, marketed as healing clay and as protein-fining agent in wine, bentonite is expected to adsorb specific compounds while having no negative effects on the quality of the product in which it is used. In this study, 34 commercially available bentonites for different applications were selected and analyzed for their elemental composition, extraction of heavy metals, swelling behavior, and protein removal rate under conditions relevant to wine. The results indicate that bentonites can have a very variable composition that does not correlate with the intended use. The extraction of heavy metals is not directly related to the raw material and depends on swelling behavior and surface area of the clay. Interestingly, there is a similar degree of variability in swelling behavior among wine bentonites as there is for healing clays. This correlates with the protein removal rate but also with the extraction of iron, a transition metal that is known for its catalytic activity for oxidation reactions. Even though the protein removal rate is much higher than for other clays, bentonites that are marketed for wine show an extraction behavior that can have a negative effect on the final product.

**Keywords:** clay; fining; extraction; heavy metals; X-ray fluorescence

## 1. Introduction

Bentonite is a montmorillonite clay with specific chemical and physical properties, making it a great tool for the removal of impurities from a multitude of products, including wine [1–3]. In slightly acidic aqueous solutions, bentonite develops a negative surface charge that, in addition to the porous clay structure, helps with removing proteins that possess a positive net charge at wine pH [4]. The use of bentonite as a protein-fining agent has a long tradition due to its low application cost and ease of use without any additional specialized equipment. There are disadvantages and drawbacks [3,5,6], however: to this date, bentonite is still the most frequently used tool to achieve protein stability in wine [7,8]. Despite a multitude of efforts to find replacements [9,10] that could match the fining performance and low cost, a real alternative has not yet been found.

Named after a rock formation near Fort Benton, Wyoming, bentonite is mined in various places around the world. While the basic elemental composition of the clay is expected to be similar, the trace elements may vary based on the location [3]. The main classifier cations that give bentonite its physical properties are sodium, calcium, potassium, or a mix of sodium and calcium. Those cations determine the swelling behavior and cation-exchange capacity of the material [3]. Other metals such as aluminum, chromium, or lead have also been identified but are considered minor components that leach out of the clay in considerably lower concentrations [11]. The application of bentonite in consumer goods that are intended for human ingestion gives rise to concerns that are related to the chemical composition of the clay. Especially in wine, which is an acidic mix of alcohol and water, the extraction conditions are ideal for heavy metals. Bentonite is added at

varying concentrations depending on—among other factors—vintage, growing conditions, and wine style [12] and usually stays in contact with the wine for multiple days [7]. If catalytic metals are extracted into the product during that time, the wine can be subject to oxidation [13] and other enzymatic and nonenzymatic degradation reactions [14]. In particular, iron and copper are a concern due to their ability to catalyze the oxidation cascade involving hydrogen peroxide [13,15].

More recently, bentonite products are marketed for their low iron content to the point where the product could be added prior to fermentation and left in contact with the wine throughout the production process [16]. This raises the question of how variable the composition of bentonites is and how much of the trace elements are extracted into the wine. The aim of this study was to analyze a large variety of wine-related bentonites and compare their characteristics to bentonites that are intended for other uses. For that comparison, an additional focus was on the desired characteristics as a wine-fining agent.

## 2. Materials and Methods

### 2.1. Bentonite Selection

The 34 bentonites included in this study were selected from a broad range of manufacturers and suppliers to cover the whole spectrum of products that are currently used in the wine industry and compare them to formulations from other areas. A total of 23 wine bentonites came from eight suppliers, 10 bentonites labeled as Healing Clay were purchased from different manufacturers, and one desiccant bentonite was added to the study as a negative control. In order to protect the identity of each product as well as the commercial interest of the manufacturers, all bentonites were anonymized and only identified with a number and their use in either wine or as a pharma/beauty product. Figure 1 shows the bentonites in their commercial form and as the pellet that was created for the X-ray fluorescence analysis.

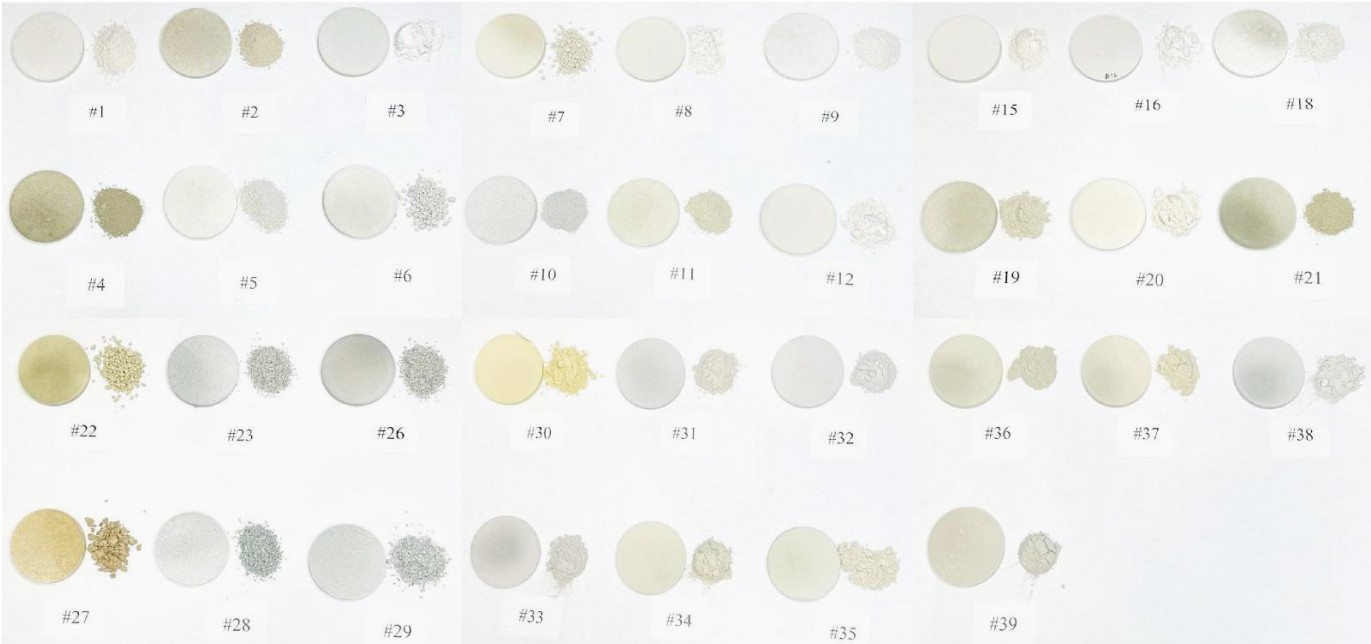

**Figure 1.** Collage of all bentonites included in this study. The left part of each product shows the pressed pellet for X-ray fluorescence analysis and the right side shows the bentonite in its commercial form.

### 2.2. X-ray Fluorescence Spectrometry

All bentonites were dried, ground into a fine powder using a pestle and mortar, mixed with Ultrabin binder (Spex SamplePrep, Metuchen, NJ, USA) at a ratio of 9+1, and pressed

into 40 mm pellets with 17 tons of pressure (Atlas Autotouch 25T, Specac Ltd., Orpington, UK) for X-ray fluorescence spectrometry (ZSX Primus IV, Rigaku Americas Corporation, Woodlands, TX, USA). Analysis conditions were used as preset in the EZ-Scan method of the ZSX Guidance software (Rigaku Americas Corporation, Woodlands, TX, USA). All elements from fluorine to uranium were screened and calculated as percent of the total dry mass.

### 2.3. Swelling Behavior and Extraction Experiments

In order to evaluate the behavior of the bentonites in aqueous suspension, 1 g of the products in their commercial form were added to a 15 mL centrifuge tube and filled to the 10 mL mark with a 1% citric acid solution (made from citric acid in water, all supplies from VWR International, Radnor, PA, USA). The tubes were closed and placed on a shaker table for 2 days allowing for complete hydration and swelling of the clay. After that, the tubes were stored upright in the fridge to settle the bentonites and the final volume was recorded after 48 h. The tubes were then mixed again and placed back onto the shaker table where the bentonites were further extracted for a total duration of 7 days. The tubes were then centrifuged at $6000 \times g$ RPM (3750 RCF) for 10 min and decanted. The clear liquid extract was stored in the refrigerator for extracted-ion analysis.

### 2.4. Microwave Plasma Atomic Emission Spectroscopy

Selected elements were analyzed in the aforementioned citric acid extract using Microwave Plasma Atomic Emission Spectroscopy (Agilent 4200 MP-AES, Agilent Technologies, Inc., Santa Clara, CA, USA). Relevant elements were selected based on the X-ray fluorescence spectrometry data of the dry bentonites and calibrated with pure standards (VeriSpec® Multi-Element Standard 18, ICCA Chemical Company, Arlington, TX, USA; all other elements from SPEX CertiPrep, Metuchen, NJ, USA). The detection wavelengths were 437.923 nm for V, 616.217 nm for Ca, 371.993 nm for Fe, 553.548 nm for Ba, 341.476 nm for Ni, 345.351 nm for Co, 405.781 nm for Pb, 766.491 nm for K, 403.307 nm for Mn, 518.360 nm for Mg, 588.995 nm for Na, and 396.152 nm for Al. The read time per element was set to three seconds with a nebulizer flow between 0.45 L/min and 0.95 L/min, depending on the element. The extraction data were calculated as mg/L and % extracted based on the XRF dry-material data.

### 2.5. Protein Removal Rate

In order to test the protein removal rate of all bentonites under conditions relevant to wine, a model wine system with 1 g/L egg-white protein as previously described by Sommer et al. [9] was used. The experiments were performed in 50 mL centrifuge tubes in duplicates and the bentonites were added at a rate of 1 g/L. The tubes were placed on a shaker table for 10 h overnight and then clarified by centrifugation. The residual protein concentration after bentonite removal was analyzed using the colorimetric assay according to Bradford (1976) [17].

### 2.6. Data Analysis

Data analysis and statistics were performed using SigmaPlot 14.0 (Systat Software, Inc., Chicago, IL, USA) and XLStat 2021 (Addinsoft, New York, NY, USA).

## 3. Results and Discussion

The bentonites selected for this study display a wide variety of colors and textures (Figure 1) independent of the intended use of the product. Since the geographical origin of the raw clays is not advertised by the manufacturers, it remains unknown if the colors indicate a specific elemental composition based on the area of origin as suggested in the literature [18]. The color, however, does not correlate with the intended use, since a similar proportion of white, grey, or brown bentonites are marketed for their use in wine or facial masks. In order to clarify if a specific elemental composition is indicative of the suggested

use, the dry bentonites were analyzed via X-ray fluorescence spectrometry. Figure 2 shows the results as a principal component analysis (PCA). This method allows for a visual correlation analysis since it groups samples based on similarities in the quantitative data and also shows positive and negative correlations of the elemental composition within the bentonites. Vectors pointing in the same direction imply a positive correlation, opposite directions describe a negative correlation, and a 90-degree angle indicates no correlation at all.

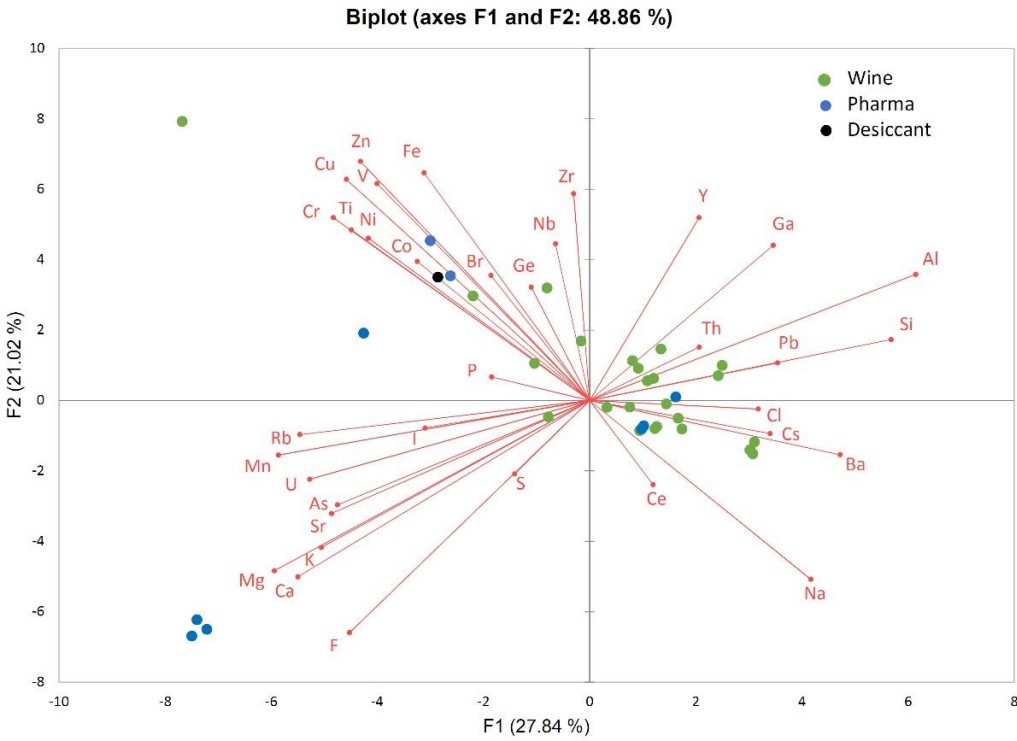

**Figure 2.** Principal component analysis of the elemental composition of the dry bentonites. Samples are grouped by intended use as wine-fining agent, pharma/beauty product, or desiccant.

Interestingly, the bentonites cannot be distinctly grouped based on their intended use. The assumption that only bentonites of a specific origin and composition would be used in a product such as wine that is made for human consumption is not supported by the data. There is a larger grouping of wine-related bentonites around the center of the graph, leaning towards aluminum and sodium, but the group is not distinctly different from the healing clay products. The vectors that describe the elements found in bentonite are also spread fairly equally around the axes, which is an indication that the variability between products is fairly large and most likely based on the geographic origin and the pretreatment by different manufacturers. This observation is supported by previous studies [19–22] that show an elemental fingerprint of clay based on geographic origin among other factors.

Even though the method of analysis via X-ray fluorescence spectroscopy is only considered semiquantitative, the method demonstrates the presence of heavy metals in most of these bentonites. However, this observation alone does not mean that those compounds are extractable into wine and could be a concern there. Wine is a high-ethanol and low-pH environment with a fairly significant redox potential, which makes it a great solvent for the extraction of ionic substances from a variety of materials. In order to identify any existing correlations between the composition of the dry clay material and the extraction of metal cations into wine, Figure 3 shows PCAs of the liquid extract expressed via a total concentration and the calculated extraction percentage between solid and liquid phase.

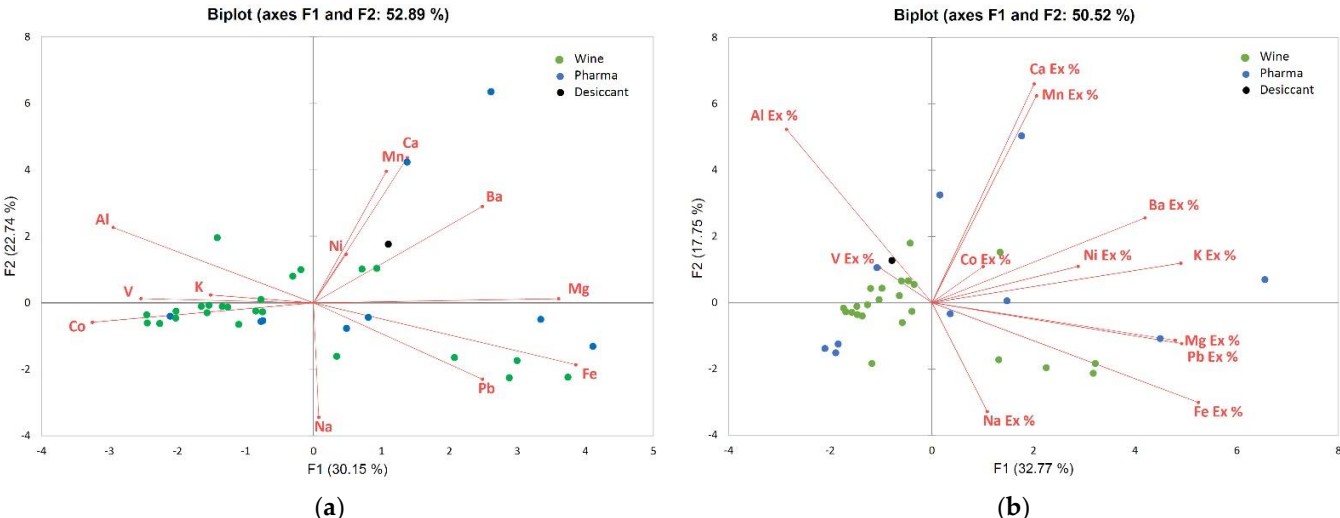

**Figure 3.** Principal component analyses of bentonite extracts, expressed as elements extracted (**a**) and percent extracted based on the analysis of the dry bentonites (**b**). Samples are grouped by intended use as wine-fining agent, pharma/beauty product, or desiccant.

Similarly to the results of the dry bentonites, there is no clear separation between wine-related products and healing clays in the acidic extracts. Interestingly, the extraction patterns of single elements do not correlate with the composition in the raw material. For example, there is no correlation between iron and lead in dry bentonites, but the two elements extract in the same samples in aqueous solution. In addition, while there is no correlation between iron and aluminum in bentonite, they show a negative correlation in solution. The reasons for this inconsistency remain unclear; however, the data indicates that the presence of potentially problematic elements in bentonite does not necessarily lead to their extraction into wine.

One reason for the low extractability of heavy metals from bentonite could be the bound form in the clay [22] and the limited surface area due to the granular structure of some of the products. The functionality as a protein-binding additive in wine, for example, is based on the increase in surface area due to the swelling behavior in an aqueous environment [4]. This behavior was tested with a 1% citric acid solution and is shown in Figure 4. All bentonites were used in their commercial form, which is an important detail since some preparations are sold as a fine powder while others are used in a granular form. The desiccant bentonite (#27) serves as a negative control here due to the gravel-like texture, which does not show any physical swelling in water. In fact, as the only product, 1 g of this bentonite retains its volume of approximately 2 mL throughout the experiment.

As reported in the literature, the swelling behavior depends on the surface area and the exchange ion of the clay [4,23]. The data of this study support this assumption. Bentonites that are sold as a fine powder occupy the largest volume after 48 h, especially if one of the exchange ions is sodium (correlation coefficient 0.485). For applications in wine, the use of mixed-cation bentonite is common [5], where sodium is paired with calcium. This combines good swelling behavior with good exchange capacity and minimal sensory impact on the product [4]. However, a larger surface area due to increased swelling is also moderately positively correlated with elevated extraction of mineral ions (correlation coefficient 0.520). This presents the challenge of optimizing the formulation of bentonites as a fining agent in wine to find the sweet spot between necessary surface area and low extraction of potentially undesirable catalytic metal ions. The extraction of iron as an oxidation catalyst can be especially problematic [24].

In order to evaluate the performance as a protein-fining agent in wine, each bentonite was used in a model system under the same conditions. Figure 5 shows the residual concentration of egg-white protein after treatment with 0.1% bentonite.

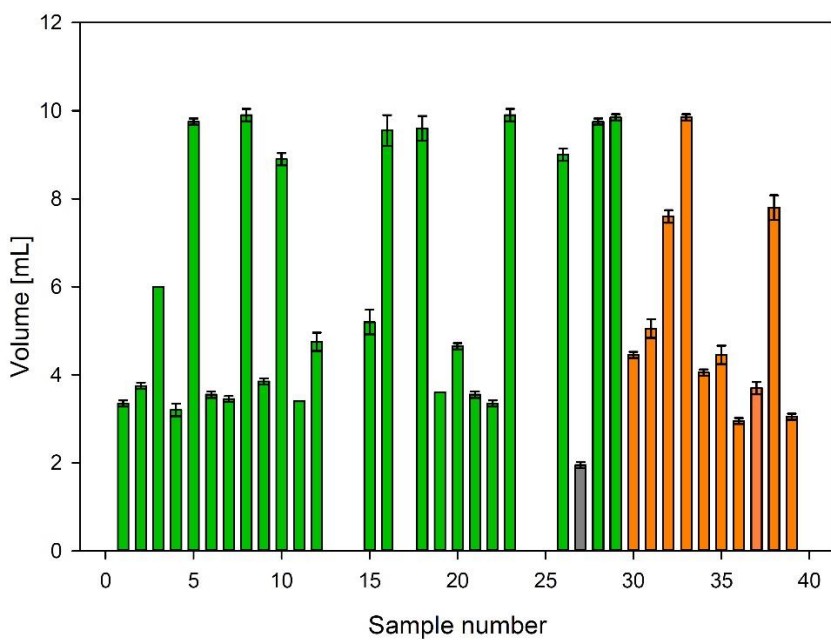

**Figure 4.** Volume of 1 g of bentonite after 48 h in a 1% citric acid solution. Green bars represent wine-fining agents, orange bars are pharma/beauty products, and the grey bar is the desiccant. Missing bars indicate unavailable products that were not included in the study.

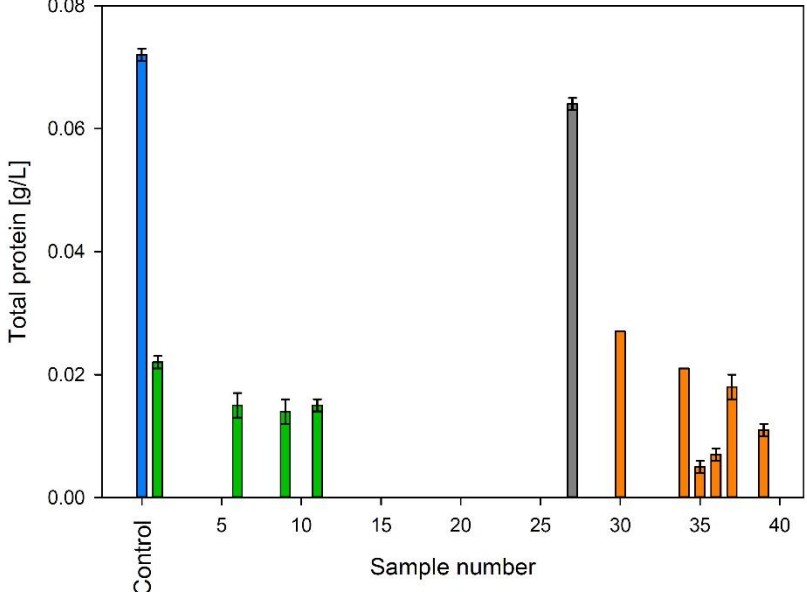

**Figure 5.** Residual protein concentration after the addition of 1 g/L bentonite to a model system. Green bars represent wine-fining agents, orange bars are pharma/beauty products, and the grey bar is the desiccant. Missing bars indicate either unavailable products that were not included in the study or bentonites that removed the protein below the detection limit.

Interestingly, the majority of bentonite clays led to a reduction in protein below the detection limit of 5 mg/L. Wine-related bentonites were more effective, with only 17% of the products leaving protein in the model wine, while 60% of healing clays did not remove all egg-white proteins from solution. The desiccant bentonite did not show any fining activity compared to the control. This illustrates that the pretreatment of bentonite clays is an important step for their functionality. While the majority of healing clays fall into the category of calcium bentonites, most wine bentonites also contain sodium as an exchange ion. The difference in swelling behavior and consequently protein-binding capacity is the

main factor that differentiates the bentonites in terms of their fining efficiency (correlation coefficient −0.933). In order to evaluate the main correlations identified in this study, Table 1 indicates the coefficients as well as a classification as weak, moderate, or strong.

**Table 1.** Summary of correlation coefficients identified in this study.

|  |  |  |
|---|---|---|
| Swelling volume [mL] | Residual protein [g/L] | −0.480 |
|  | Average ionic extraction [%] | 0.365 |
|  | Iron extraction [%] | 0.613 |
|  | Exchange ion Ca | −0.455 |
|  | Exchange ion Na | 0.485 |
| Wine bentonite | Exchange ions Na and Ca | 0.415 |
|  | Other uses (healing clay) | −0.933 |
| Iron extraction [%] | Average extraction [%] | 0.520 |
|  | Exchange ion Ca | −0.438 |
|  | Exchange ion unknown | 0.357 |

>0.3: weak correlation; >0.5: moderate correlation; >0.7: strong correlation.

As discussed above, the swelling behavior of bentonite is related to the most relevant attributes for its use in wine. It is determined by the exchange ion and defines all extraction and adsorption properties of the clay. The exchange ion was not defined in very few bentonites; however, this group of products only shows a weak positive correlation with total iron extraction.

## 4. Conclusions

The variability of bentonites in regard to color and texture is substantial but does not correlate to a specific use. Commercial bentonites that are intended for different applications show a very diverse composition that seems to be less related to their use but more to their geographic origin. The extraction of potentially problematic elements from the clay depends on the pretreatment and the specific formulation that influences the swelling behavior. The swelling volume is the single most important attribute for bentonite use in wine, since it influences protein-removal behavior and iron extraction into the beverage, which can lead to undesirable effects such as oxidation and accelerated aging. Even bentonites that are marketed to have low iron content can still release it into the wine. Bentonites that are intended for uses other than wine have the ability to bind protein, but are less effective on average.

**Author Contributions:** Conceptualization, S.S.; methodology, S.J.S. and S.S.; formal analysis, M.G., S.J.S. and S.S.; investigation, M.G., S.J.S. and S.S.; data curation, S.S.; writing—original draft preparation, S.S.; writing—review and editing, S.S.; visualization, S.S.; project administration, S.S. All authors have read and agreed to the published version of the manuscript.

**Funding:** This research received no external funding.

**Institutional Review Board Statement:** Not applicable.

**Informed Consent Statement:** Not applicable.

**Acknowledgments:** We would like to thank Benjamin Andersen in the Department of Earth and Environmental Sciences at Fresno State for his guidance and support with the X-ray fluorescence spectrometry.

**Conflicts of Interest:** The authors declare no conflict of interest.

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
