# Peer review of "Characterization of Different Bentonites and Their Properties as a Protein-Fining Agent in Wine"

_beverages, doi:10.3390/beverages8020031_

Round 1

Reviewer 1 Report

The manuscript of the article deals very thoroughly with the characterization of 34 different commercial bentonites and their basic cleaning properties of protein removal in wine. Many of the obtained results are very clearly shown, which are statistically processed accordingly. The results shown in 5 figures and 1 table are followed by a quality discussion and an important conclusion of the research, and appropriate references are used.
I have no comments or corrections, so there is no attachment.

Author Response

We would like to thank the reviewer for performing the review and the kind comments. We appreciate the encouraging remarks.

Reviewer 2 Report

The publication deals with the topic of the use of bentonite in winemaking. The discussed topic is an important in wine production, known for many years, however, the proposed research in the full scope of the work highlights its advantages and disadvantages as well as possible applications in this field. Reliable analyzes were used, the results of the research were correctly summarized in graphical form and the results were statistically processed. The study fully reflects the results obtained. The research results were summarized in a synthetic way. The only suggestion of the reviewer is 21 items of the cited literature, they seem to be a bit limited, the more that only 7 items are items from the last 12 years, and the rest are much older. In the opinion of the reviewer, the cited literature should also be updated

Author Response

We would like to thank the reviewer for the constructive criticism. All changes are marked in red in the revised manuscript.

We have added 3 newer references to the manuscript to update the literature. The older references were kept in order to give credit to the original research which was done in the 80s and 90s. We consider this important for the scientific value of the manuscript.

Reviewer 3 Report

  • The research studied the properties of bentonite to be use as fining agent in wine: the are compared with other bentonites used in pharma.
  • Characterization of bentonites and its activity as protein removal is very relevant in winemaking because high doses of bentonite diminish quality of wine. Due climate change, protein instability is higher each harvest and higher doses of bentonite are used. As this paper explains, to use these high doses of bentonite, has consequences in terms of wine quality. To know the characteristics of different bentonites and which is more effective could be very helpful. So, the relevance of this paper is high from the oenological point of view.
  • Methodology used is adequate and gives enough information for bentonite characterization, no just from chemical point of view, but enological side.
  • I found no clear the comment in line 157 about Figure 3, neither the figure. Please explain further.: To correlate the composition of the raw clay material with the expected extraction in wine, Figure 3 shows 157 PCAs of the extract expressed as a total concentration and the calculated extraction per-158 centage. 159
  • The paper well written and the text is clear. Figures help text comprehension .
  • Conclusions are consistent and they address the main objective of the paper. It is concluded that the swelling volume is the single most important attribute for bentonite use in wine, since it influences protein removal behavior and iron extraction into the beverage, which can lead to undesirable effects like oxidation and accelerated aging, which is relevant for wine making.

Author Response

We would like to thank the reviewer for the constructive criticism and helpful comments. All changes are marked in red in the manuscript.

We rephrased the statement in line 157 and added an explanation.